Immunohistochemistry for the detection of neural and inflammatory cells in equine brain tissue

Delcambre Gretchen H. 1 gh.delcambre@colostate.edu
Liu Junjie 2
Herrington Jenna M. 2
Vallario Kelsey 2
Long Maureen T. 2
1 Department of Biomedical Sciencess/College of Veterinary Medicine and Biomedical Sciences, Colorado State University , Fort Collins, Colorado , USA
2 Department of Infectious Diseases and Pathology/College of Veterinary Medicine, University of Florida , Gainesville, Florida , USA
Esteban María Ángeles
Electronic publication date: 2016 Jan 25
Publication date: 2016
Volume: 4
Electronic Location ID: e1601
Received 2015 Oct 2; Accepted 2015 Dec 23
Copyright: © 2016 Delcambre et al.
Copyright year: 2016
Copyright holder: Delcambre et al.
License: This is an open access article distributed under the terms of the Creative Commons Attribution License, which permits unrestricted use, distribution, reproduction and adaptation in any medium and for any purpose provided that it is properly attributed. For attribution, the original author(s), title, publication source (PeerJ) and either DOI or URL of the article must be cited.
License URL: https://creativecommons.org/licenses/by/4.0/

Keywords: Immunohistochemistry, Equine, Neuropathology, Leukocytes, Neuroglia

Funding: This work was funded by the Fern Audette Endowment for Equine Studies. The funders had no role in study design, data collection and analysis, decision to publish, or preparation of the manuscript.

==============================
Phenotypic characterization of cellular responses in equine infectious encephalitides has had limited description of both peripheral and resident cell populations in central nervous system (CNS) tissues due to limited species-specific reagents that react with formalin-fixed, paraffin embedded tissue (FFPE). This study identified a set of antibodies for investigating the immunopathology of infectious CNS diseases in horses. Multiple commercially available staining reagents and antibodies derived from antigens of various species for manual immunohistochemistry (IHC) were screened. Several techniques and reagents for heat-induced antigen retrieval, non-specific protein blocking, endogenous peroxidase blocking, and visualization-detection systems were tested during IHC protocol development. Boiling of slides in a low pH, citrate-based buffer solution in a double-boiler system was most consistent for epitope retrieval. Pressure-cooking, microwaving, high pH buffers, and proteinase K solutions often resulted in tissue disruption or no reactivity. Optimal blocking reagents and concentrations of each working antibody were determined. Ultimately, a set of monoclonal (mAb) and polyclonal antibodies (pAb) were identified for CD3+ (pAb A0452, Dako) T-lymphocytes, CD79αcy+ B-lymphocytes (mAb HM57, Dako), macrophages (mAb MAC387, Leica), NF-H+ neurons (mAb NAP4, EnCor Biotechnology), microglia/macrophage (pAb Iba-1, Wako), and GFAP+ astrocytes (mAb 5C10, EnCor Biotechnology). In paraffin embedded tissues, mAbs and pAbs derived from human and swine antigens were very successful at binding equine tissue targets. Individual, optimized protocols are provided for each positively reactive antibody for analyzing equine neuroinflammatory disease histopathology.

Introduction

Comparative pathology of many diseases that affect multiple species is hindered by the lack of species-specific reagents. Although many antibodies that react with cellular antigens and inflammatory molecules are available, there still are gaps in how these work in formalin-fixed paraffin embedded (FFPE) tissues. This limits our ability to generate new knowledge about disease processes that can benefit human and animal health as a whole. Horses develop many of the same or similar central nervous system (CNS) infections as humans (Ritchey et al., 2006; Bourgeois et al., 2011; Rushton et al., 2013; Yu et al., 2015). Cell marker panels are often composed of both equine and non-equine specific antibodies, of which most are used in flow cytometry. Since manufacturers may not have supporting technical documentation on whether their products will cross-react with equine antigens (Beckstead, 1994; Ramos-Vara, 2005), development of antibody panels to accomplish in situ disease characterization in formalin-fixed tissue is a formidable task because cross-linking of antigens often renders epitopes non-reactive. Additional investigative steps are required to determine if antigens can be retrieved and what retrieval method works best. Because of this and other hurdles, there is limited information on what antibodies react with FFPE tissue.

The goal of this study was to identify a panel of cell markers for studying cellular pathogenesis in equine infectious brain diseases. In particular, protocols for phenotyping reactive glia, neurons, and infiltrating peripheral blood mononuclear and polymorphonuclear cells in the equine brain were developed. Commercially available antibodies for CD3+ T-lymphocytes, CD8+ and CD4+ T cell subpopulations, B lymphocytes, macrophages, microglia, astrocytes, and neurons were investigated for reactivity in normal and diseased FFPE tissue. Optimization of manual immunohistochemistry (IHC) protocols was performed using various IHC staining reagents and methods.

Materials and Methods

Tissue samples

Immunohistochemistry protocols were developed on diseased and normal horse tissues. Diseased horses consisted of clinically affected West Nile virus (WNV), both naturally and experimentally infected, and Sarcocystis neurona infections (Beckstead, 1994; Gutierrez et al., 1999; Porter et al., 2003; Seino et al., 2007). Neurologically, normal horses were obtained by owner surrender for humane euthanasia due to loss of use. Brain, spinal cord, lymph node, spleen, thymus, and liver were collected and archived from these animals under University of Florida (Gainesville, FL) Institutional Animal Care and Use Committee protocols #F077, #F093, #D163, and #4109. Tissues were fixed in 10% neutral buffered formalin and processed into paraffin-embedded blocks approximately one-week after fixation. Initial evaluation of antibody binding was tested on non-infected equine lymph node and spleen for lymphocytic targets, liver and thymus for tissue macrophage targets, and brain for microglia, astrocytes, and neurons.

Tissue processing

The invariable IHC procedures for all protocols included sectioning FFPE tissues at 5 μm and placing them on positively charged glass slides. The slides were soaked in xylene (Fischer Scientific, Pittsburg, PA, USA) three times for 5 min to remove paraffin. These sections were then rehydrated through a gradient of ethanol (Fischer Scientific) for 5 min in each concentration, 100%, 100%, 95%, and 70% ethanol, followed by de-ionized water. In order to reduce the volume of the reagents tested and liquid loss, tissues were encircled with a hydrophobic barrier pen (ImmEdge™ Pen, Ted Pella Inc., Redding, CA, USA).

Antigen unmasking

Three methods of heating slides for investigating heat induced epitope retrieval (HIER) effectiveness included using a pressure cooker, microwave, and a double boiler. Pressure cooking was performed at 125 °C for 30 s followed by 90 °C for 10 s (Matyjaszek et al., 2009; Grosche et al., 2012), or microwaving was performed for 10 min (Kumar & Rudbeck, 2009) using a commercial countertop GE 1000W oven. For double boiling, two tissue-slides were floated back-to-back in 25 ml of retrieval solution in a 50 ml plastic conical tube. Conical tubes were then placed in pre-warmed water of a 250 ml glass beaker on a hotplate. Water temperature was maintained at 90 °C. After approximately 5 min of warming the retrieval solution, HIER was timed for 10 min. Conical tubes were then removed from the double boiler and allowed to cool for 15 min at 27 °C. Tissues were rinsed in deionized water three times for 2 min.

Heat induced epitope retrieval buffers were tested primarily using the double boiler system. Regents included two commercial citrate buffers, Epitope Retrieval Solution pH 6 (Novacastra, Leica, Newcastle Upon Tyne, UK) and Target Retrieval Solution pH 6 (Dako, Glostrup, Denmark), and an ethylenediaminetetraacetic acid (EDTA) solution buffered at pH 9 (10 mM Tris Base, 1 mM EDTA solution, 0.05% Tween 20, and NaOH to titrate to pH 9). A 1 × concentration of each solution was freshly made by diluting stock solutions with deionized water. For proteolytic epitope retrieval, tissues were treated with 200 ug/ml proteinase K solution (Tris HCL 100 mM pH 8.2, Tween 20, and Proteinase K (Ambion, Foster City, CA, USA)) for 10 min at 37 °C.

Endogenous peroxidase blocking

Peroxidase neutralizing solutions that were tested included two prepared solutions of hydrogen peroxide (H2O2), 3% and 0.3% H2O2, and a ready-to-use commercial reagent, Peroxidase Block (Novolink™ Polymer Detection System; Leica, Wetzlar, Germany). Solutions containing 3% and 0.3% H2O2 were made fresh for each staining attempt by diluting 30% H2O2 (Fischer Scientific) in 1 × phosphate buffered saline (PBS) (10 × PBS, Fischer Scientific). Tissues were immersed in peroxidase blocking solution for 5 min followed by two, 5 min rinses in PBS.

Non-specific protein blocking

Non-specific blocking techniques included four commercial reagents and one lab prepared solution. Commercial reagents included 10% Normal Goat Serum (Invitrogen, Frederick, MD, USA), Protein Block (Novolink™ Polymer Detection Systems; Leica, Wetzlar, Germany), Novocastra™ IHC/ISH Super Blocking Solution (Leica), and Novocastra™ Liquid Serum, Normal Goat Serum Blocking Reagent (Leica, Wetzlar, Germany). Additionally, a 5% goat serum solution was prepared by diluting Immunopure® Goat Serum (ThermoFischer Scientific, Waltham, MA, USA) in 1 × PBS. Protein blocking reagents were applied for 20 min with the exception of Novolink™ Protein block for only 5 min. Blocking reagents were removed without rinsing before adding primary antibody.

Primary antibodies

Primary antibodies tested targeted cell populations of astrocytes, microglia, neurons, T lymphocytes, B lymphocytes, and macrophages (Table 1). All primary antibodies were diluted in one of two commercial diluents (IHC Diluent (Novacastra, Leica); Dako Antibody Diluent (Dako, Glostrup, Denmark)). Two-fold serial dilutions of each antibody were tested to determine an optimal staining range. If the signal was weak to absent or background staining was present, additional dilution tests were performed until optimal staining was achieved. Antibodies were applied in a 37 °C, humidified incubator for 60–120 min or overnight (∼16 hrs) at 4 °C in a humidified, covered dish. After primary antibody incubation, slides were washed three times for 5 min each in 1 × PBS before secondary antibody application.

Table 1 Antibodies tested for immunohistochemical reactivity in equine tissues.

Antibody	Antigen species	Host species	Isotype	Name or clone	Protein target	Source	
CD3	Human	Mouse	IgG1	F7.2.38	Thymocytes, T lymphocytes, natural killer cells	Dako	
CD3	Human	Rabbit	–	A0452	T lymphocytes	Dako	
CD4	Equine	Mouse	IgG1	CVS4	T helpers	AbD	
CD4	Human	Mouse	IgG1	4B12	Thymocytes, T helpers, mature peripheral T cells, TReg or cytotoxic T cell subsets	Dako	
CD4	Human	Mouse	IgG1	1F6	T helpers	Leica	
CD4	Equine	Mouse	IgG1	HB61A	Thymocytes, T helpers	VMRD	
CD8	Equine	Mouse	IgG1	CVS8	Cytotoxic T lymphocytes	AbD	
CD8	Equine	Mouse	IgG1	HT14A	Thymocytes, Cytotoxic T lymphocytes	VMRD	
CD5	Equine/Bovine	Mouse	IgG2A	B29A	B lymphocytes only, in horses	VMRD	
CD20	Human	Mouse	IgG1	MJ1	B lymphocytes	Leica	
CD21	Human	Mouse	IgG1	Bu33	B lymphocytes	AbD	
CD21	Human	Mouse	IgG1	1F8	B lymphocytes	Dako	
CD79αcy	Human	Mouse	IgG1	HM57	B lymphocytes	Dako	
IgG (H&L)	Equine	Goat	IgG	A70–106	B lymphocytes	Bethyl	
IgG (H&L)	Equine	Rabbit	IgG	A70–118	B lymphocytes	Bethyl	
IgG (H&L)	Goat	Rabbit	IgG	A50–100	B lymphocytes	Zymed	
Macrophage	Rabbit	Mouse	IgG1	RAM11	Macrophages	Dako	
Macrophage	Human	Mouse	IgG1	MAC387	Macrophages	Leica	
Macrophage	Human	Mouse	IgG1	AM-3K	Macrophages	TransGenic	
CD68	Human	Mouse	IgG1	CD68-KP1	Macrophages	Leica	
NF-H	Bovine	Mouse	IgG2b	9B12	Neurofilament-Heavy chain	EnCor-Biotechnology	
NF-H	Bovine	Mouse	IgG1	AH1	Neurofilament-Heavy chain	EnCor-Biotechnology	
NF-H	Swine	Mouse	IgG1	NAP4	Neurofilament-Heavy chain	EnCor-Biotechnology	
GFAP	Swine	Mouse	IgG1	5C10	Astrocytes, Satellite cells, Schwann cells	EnCor-Biotechnology	
Microglia	Human	Rabbit	–	Iba-1	Microglia, macrophages	Wako	
Neu-N	Mouse	Mouse	IgG1	A60	Neurons	Millipore	

Negative controls for each primary antibody consisted of either an isotype-matched negative primary control (MCA928, AbD Serotec, Kidlington, UK) for monoclonal antibodies or rabbit serum for polyclonal antibodies.

Detection systems

Three commercial, horseradish peroxidase (HRP)-linked conjugate detection kits were utilized by following manufacturer’s instructions. These kits included the Vectastain® ABC Kit–Mouse IgG (Vector Laboratories, Burlingame, CA, USA), the Vectastain® ABC Kit–Rabbit IgG, and the Novolink™ Polymer Detection System (Leica, Wetzlar, Germany).

Lastly, a substrate-chromogen (Vector NovaRED Peroxidase Substrate, Vector Laboratories) was applied for 10 min, rinsed with de-ionized water for 5 min, counterstained with Lab Vision™ Mayer’s Hematoxylin (ThermoFisher Scientific) for 1 min, and rinsed in running tap water for 2 min. All sections were then dehydrated through an increasing gradient of ethanol, 50%, 70%, 95%, and 100%, for 2 min each. Slides were cleared in xylene three times for 3 min each before coverslipping with Permount™ mounting medium (Fischer Scientific, Hampton, NH, USA).

Results

Antigen unmasking

Of the reagents tested, low pH citrate based solutions resulted in superior staining (Table 2). Target Retrieval Solution pH 6 by Dako was sufficient for most antibodies; however, use of Epitope Retrieval Solution pH 6 by Leica resulted in positive staining when the Dako product failed to produce staining with the CD79αy+ antibody. Proteinase K solution was tested with macrophage antibodies and resulted in increased staining intensity of MAC387+ antibody.

Table 2 Selected protocols for immunohistochemical visualization of resident and infiltrative cells in normal and WNV-diseased equine CNS tissues.

Antibody	ID/clone	Tissue format	Peroxidase blocker	Antigen retrieval	Blocker	Dilution	Incubation	Commercial secondary kit	
CD3	A0452	FFPE	3% H2O2	Target retrieval solution pH65	10% normal goat serum1	1:100	60 min at 37 °C	Vectastain® ABC Kit–Rabbit3	
CD4	HB61A	FFT	0.3% H2O2	None	Protein block2	1:800	120 min at 23 °C	Post primary block & Novolink polymer2	
CD8	HT14A	FFT	0.3% H2O2	None	Protein block2	1:800	120 min at 23 °C	Post primary block & Novolink polymer2	
CD79αCY	HM57	FFPE	3% H2O2	Epitope retrieval solution pH62	5% immune pure goat serum4	1:100	90 min 37 °C	Post primary block & Novolink polymer2	
Macrophage	MAC387	FFPE	3% H2O2	Target retrieval solution pH65 or proteinase K	10% normal goat serum1	1:100	60 min at 37 °C	Vectastain® ABC Kit–Mouse3	
NF-H	NAP4, AH1, 9B12	FFPE	3% H2O2	Epitope retrieval solution pH62	Protein block2	1:1,000	60 min at 37 °C	Post primary block & Novolink polymer2	
GFAP	5C10	FFPE	3% H2O2	Target retrieval solution pH65	5% immune pure goat serum4	1:8,000	60 min at 37 °C	Vectastain® ABC Kit–Mouse3	
Iba-1	–	FFPE	3% H2O2	None	Protein block2	1:500	60 min at 37 °C	Post primary block & Novolink polymer2	
Notes:

FFPE, Formalin-fixed, paraffin embedded; FFT, Fresh, frozen tissue.

1 Invitrogen™.

2 Leica®–Novocastra™ Product Line.

3 Vector Labs®.

4 FischerScientific®.

5 Dako®.

Various issues were encountered with the method of applying heat. Utilizing, one antibody, CD79αy+, only the double boiling method resulted in consistent positive staining of lymph nodes. Pressure cooking and microwaving resulted in limited to no staining or uneven staining, respectively. In addition, tissue disruption was minimal with the double boiling method. One antibody, Iba-1+, did not require heat retrieval and any HIER resulted in non-specific background staining.

Endogenous peroxidase blocking

In FFPE tissues, 3% H2O2 solution was reliable for reducing background staining without disruption of the tissue. There was no difference between commercial and lab diluted reagents.

Non-specific protein blocking

Multiple combinations of IHC antibody diluents and non-specific protein blocking reagents were tested. No difference in staining intensity was noted between of either diluent used. Among protein blocking solutions, 10% Normal Goat Serum, Protein Block from the Novolink™ Polymer Detection System, and 5% Immunopure® Goat Serum in PBS solution were all equal in blocking non-specific staining.

Specificity and sensitivity of monoclonal and polyclonal antibodies

Several antibodies were tested for lymphocyte specific CD antigens including CD3+, CD4+, CD5+, CD8+, CD20+, CD21+, and CD79αy+ (Table 1). Of the two CD3+, pan-T cell markers examined, rabbit anti-human polyclonal antibody (A0452, Dako) appropriately stained lymph node cortex (Fig. 1A), periarterial lymphatic sheaths of spleen, and perivascular cuffs in WNV infected brain (Fig. 1B). Four CD4+ T-helper cell antibodies were tested. Only mouse, monoclonal anti-equine antibody (HB61A, VMRD, Pullman, WA, USA) at 1:25 dilution positively stained lymph node cortex; however, background staining was high when applied to brain tissues. Two CD8+ cytotoxic T cell markers were investigated, but neither marker had reactivity in FFPE tissue.

Figure 1 IHC of CD3+ T lymphocytes and CD79+ B lymphocytes in euqine tissues.

For the detection of lymphocytes, (A, C) equine lymph node cortex and (B, D) WNV infected equine brain incubated with (A, B) CD3+ T lymphocyte primary antibody (pAb A0452; Dako, Glostrup, Denmark) for 60 minutes at 37 °C and detected by Vectastain® ABC Kit, or incubated with (C, D) CD79αcy+ B lymphocyte primary antibody (mAb HM57, Dako) for 90 minutes at 37 °C and detected by Novolink™ Polymer Detection System. Vector NovaRED Peroxidase Substrate chromogen and hematoxylin couterstain. Bar, 50 μm.

Antibodies against CD5+, CD20+, CD21+, CD79αy+, and IgG (H+L) (Table 1) were tested for identification of B cell populations. A putative lymphocyte marker, CD5+ (B29A, VMRD), reportedly selects for B cells in equine tissues. Although this antibody intensely stained the germinal centers of FFPE lymph nodes, cortical staining was also noted. Because of this, CD5+ (B29A) was unreliable for distinction between B cell and T cell populations. Additionally in brain tissue, this antibody resulted in non-specific background staining, which could not be resolved. No staining was achieved with CD20+, CD21+, or any IgG (H+L) antibodies. Anti-human CD79αcy+ (HM57; Dako, Glostrup, Denmark) monoclonal antibody at 1:100 successfully stained lymph node germinal centers with no background staining in equine brain (Figs. 1C and 1D).

Multiple macrophage-targeting antibodies were investigated. RAM 11 (Dako, Glostrup, Denmark), AM-3K (TransGenic Inc., Kobe, Japan), and CD68+ (KP1; Leica, Wetzlar, Germany) antibodies had no reactivity with control tissues; however, MAC387+ (Leica, Wetzlar, Germany) was reactive. This marker positively stained control hepatic and thymic macrophages (Fig. 2A) with limited staining noted in lymph node sections. Based on cell morphology, polymorphonuclear and mononuclear cells also stained positively due to lack macrophage-specificity of the antibody. No reactivity was noted in normal equine brain. The MAC387+ cell population was distinct from the distribution of CD3+ lymphocytes in brain sections with inflammation due to WNV infection (Fig. 2B). Additionally, this macrophage antibody had little to no cross reactivity with brain microglia since few MAC387+ cells were visualized within glial nodules of WNV+ brain.

Figure 2 IHC of MAC387+ and Iba-1+ macrophages and microglia in equine tissues.

For the detection of macrophage/microglial lineage cells, (A) equine thymus and (B) WNV infected equine brain were incubated with MAC387+ primary antibody (mAb MAC387; Leica, Wetzlar, Germany) for 60 minutes at 37 °C and detected by Vectastain® ABC Kit. Additionally, (C) normal (arrows) and (D) WNV infected horse brain were incubated with microglia/macrophage primary antibody (pAb Iba-1; Wako, Neuss, Germany) for 60 minutes at 37 °C and detected by Novolink™ Polymer Detection System. Vector NovaRED Peroxidase Substrate chromogen and hematoxylin couterstain. Bar, 50 μm.

To characterize reactive gliosis, antibodies against both microglial and astrocytic markers were tested. In non-infected brain tissue, Iba-1+ antibody stained scattered resident microglia with low intensity, yet microglial processes were visualized (Fig. 2C). Staining intensity was higher in WNV-infected brains (Fig. 2D). Astrocyte populations were identified with antibody against an intermediate filament, glial fibrillary acid protein (GFAP), specific to astrocytes in CNS tissues. This antibody reacted strongly with astrocytes in both normal and infected brains with distinct astrocytic processes notable. GFAP+ astrocytes were distributed within the brain parenchyma, at the glia limitans, along brain vasculature, and in areas of gliosis of WNV+ brains (Fig. 3).

Figure 3 IHC of GFAP+ astrocytes in WNV infected equine brain.

GFAP+ astrocytes in WNV infected equine brain (A) near blood vessels and (B) at the glial limitans. IHC. Bar, 50 μm.

Three antibodies that target neurofilament heavy-chain proteins (NF-H) of neuronal axons successfully stained equine brain tissue. Swine derived, NAP4 (EnCor Biotechnology Inc., Gainesville, FL, USA) antibody, was superior and stained neurons in both normal (Fig. 4) and infected brains. Mouse monoclonal Neu-N antibody (A60, Millipore, Billerica, MA, USA), which targets neuronal perikaryon, was also tested but without successful reactivity in equine tissue.

Figure 4 IHC of NF-H+ neurons in equine brain.

Normal equine brain incubated with NF-H+ neuron primary antibody (mAb NAP4; EnCor Biotechnology, Gainesville, FL, USA), for 60 minutes at 37 °C and detected by Novolink™ Polymer Detection System. Vector NovaRED Peroxidase Substrate chromogen and hematoxylin couterstain. Bar, 50 μm.

Successful manual IHC protocols for these antibodies in equine neural tissue were identified (Table 2). It should be noted that overnight, refrigeration with CD3+, CD79+, MAC387+, GFAP+, and Iba-1+ primary antibodies will successfully stain tissues and may provide flexibility in laboratory scheduling if single-day protocol completion is not possible.

Discussion

Formalin-fixed, paraffin embedded tissues are commonly archived for histological examination. Formalin fixation and paraffin embedding samples retains tissue architecture and deactivates any potential infectious agents (Cantile et al., 2001; Ritchey et al., 2006; van Marle et al., 2007); however, this process can block or destroy sensitive epitopes (Beckstead, 1994; Mérant et al., 2003; Terio et al., 2003; Ibrahim et al., 2007). Despite reported reversal of these treatments, the tested equine antigen derived CD4+ and CD8+ T cell were not detectable in FFPE. This result has been described in other species (Beckstead, 1994; Bilzer et al., 1995; Zeng et al., 1996; Gutierrez et al., 1999; Härtig et al., 2009). Anti-CD4+ (mAb HB61A, VMRD) and CD8+ (mAb HT14A, VMRD) monoclonal antibodies, however, were reactive to fresh, frozen equine lymph node and infectious CNS tissue. Archiving tissues in both FFPE and fresh, frozen tissue formats is recommended when investigating epitopes that may be affected by fixation.

After successful staining of control tissues, normal and pathologic brain tissue samples were then tested alongside positive and negative tissue controls. Pathologic tissue samples included WNV-infected brain, which contains reactive gliosis and perivascular cuffing of inflammatory cells (Cantile et al., 2001; Ibrahim et al., 2007; van Marle et al., 2007; Schnabel et al., 2013). S. neurona infected tissues were also used, which contained focal nonsuppurative inflammation, mononuclear perivascular cuffing, and the occurrence giants cells and eosinophils (Boy, Galligan & Divers, 1990; Dubey et al., 2001). Following this workflow of tissue testing aided in the identification of successful antibody reactivity.

Several hurdles must be overcome to optimize the interaction of antibodies with their intended targets. Aldehyde cross-links that bind tissue proteins during formalin fixation process must be removed by HIER or proteolytic epitope retrieval to allow epitopes to resume a more natural confirmation and increase antibody-binding capacity (Shi, Key & Kalra, 1991; Ferrier et al., 1998; Krenacs et al., 2010). Peroxidases that naturally occur within tissues must be neutralized with H2O2 in order to prevent non-specific staining during the application of peroxidase-based substrate kits (Wendelboe & Bisgaard, 2013). Endogenous proteins may non-specifically interact with antibodies and cause background staining that masks target antigen signal (Daneshtalab, Doré & Smeda, 2010; Buchwalow et al., 2011). These unwanted binding sites must be blocked. In this study a variety of reagents and methods were tested for each individual manual IHC protocol. A base set of reliable solutions and techniques for testing antibodies was identified in this study. These included 3% H2O2, low pH, citrate based HIER reagents in a double-boiler, 5% Immunopure® Goat Serum in PBS, Leica’s IHC Diluent, antibody incubation for one hour at 37 °C, and the Novolink™ Polymer Detection System. However, with meticulous tailoring of each antibody protocol, our results contained a variety of reagents that were ultimately selected for each staining procedure.

Identification of equine-reactive antibodies can be challenging due to limited knowledge of commercial antibodies’ reactivity with veterinary tissues and lack of development of multiple species-specific reagents. Evaluation of cross-species reactivity of commercially available antibodies, particularly CD antigens, has been largely performed in equine tissues prepared for whole-cell analysis like flow-cytometry (Johne et al., 1997; Mérant et al., 2003; Terio et al., 2003; Kunisch et al., 2004; Ibrahim et al., 2007) or on fresh or frozen specimens (Bilzer et al., 1995; Zeng et al., 1996; Lemos et al., 2008; Härtig et al., 2009). Large screenings of non-equine derived antibodies have often resulted in limited identification of equine reactive reagents (Ibrahim et al., 2007; Schnabel et al., 2013; Szabo & Gulya, 2013). Of the 26 antibodies in this study, six antibodies successfully reacted with FFPE equine tissues. All six antibodies were derived from non-equine antigens. Their reactivity to equine tissue is likely due to conserved epitope targets and the intracellular location of the antigen peptides (Jones et al., 1993; Ahmed et al., 2007). If an antigen’s peptide sequence is highly conserved across multiple species, the production of antibodies from the immunized host may be inhibited due to immune tolerance. However, because these antigens have intracellular origins, the host (i.e. rabbit or mouse) is more likely to elicit an immune response upon antigen exposure. In this way, these antibodies against human or swine antigen may successfully be produced and react with equine or other species’ tissues.

Commercial macrophage antibodies are non-specific to macrophages and often cross react with monocytes, granulocytes, dendritic cells, and fibroblasts (Johne et al., 1997; Kunisch et al., 2004; Shaw et al., 2005; Sellner et al., 2014). Multiple macrophage-directed antibodies with different target antigens were tested so that a distinction could be made between tissue macrophage populations and cells that may cross-react with those antibodies. RAM11 (Dako, Glostrup, Denmark) recognizes an uncharacterized, cytoplasmic antigen specific to rabbit macrophages. AM-3K (TransGenic, Strasborg, France) anti-macrophage antibody raised against human alveolar macrophage antigen recognizes cytoplasmic membrane epitopes. Anti-CD68+ (KP1; Leica, Wetzlar, Germany) recognizes primarily lysosomal membrane proteins of macrophages, secondarily macrophage membranes, and is found on monocytes, neutrophils, basophils, and large lymphocytes. None of these three antibodies were successfully reactive in FFPE brain or lymphoid tissues. Although frequently used as a macrophage-characterizing marker, MAC387+ (Leica, Wetzlar, Germany) recognizes cytoplasmic, human leukocyte antigen, L1 protein found on neutrophils, monocytes, and certain reactive macrophages. Only MAC387+ antibody was reactive with FFPE equine tissues, so the positively stained population must be considered as a mix of both macrophages and neutrophils. This cell population had a distinct distribution in serial sections of WNV-infected brain each immunolabeled for MAC387+, CD3+ T lymphocytes, and microglia. In S. neurona infected equine tissues, where multi-nucleated giant cells are observed, MAC387+ antibody did not react with these cells even though there was reaction with single cells within each lesion.

Microglia, which are of a monocytic lineage, were identified with anti-Iba-1-antibody (Wako, Neuss, Germany). This antibody recognizes an intracellular, calcium binding protein basally expressed by both microglia and macrophages, and upregulated in microglia following injury. Cell morphology, specifically microglial processes, may be used to manually separate microglia and macrophage populations. A limited number of equine studies have investigated microglial activation using major histocompatibility complex (MHC) markers (Mullen, Buck & Smith, 1992; Lemos et al., 2008). MHC predominantly targets activated microglia populations, whereas anti-Iba-1+ antibody will stain microglia in both their resting and activated form (Szabo & Gulya, 2013). Additionally, using MHC as a marker for microglial cells in an inflammatory disease would also identify all peripheral and CNS cells that have MHC upregulated due to this biological process. Characterization of the total, both resting and activated, microglia population is therefore improved with the use of Iba-1+ antibody (Ahmed et al., 2007).

Regarding detection of neurons, it must be recognized that NF-H proteins are concentrated within axons and are weakly present within cell bodies. Changes in staining intensity within cell bodies and dendrites with NF-H antibodies, such as NAP4 that targets phosphorylated NF-H proteins, are useful for identifying neuropathology (Shaw et al., 2005; Sellner et al., 2014). For quantification of neuronal populations, an antibody specific to neuronal nuclei should be used. Neu-N antibodies, which target DNA-binding proteins of neural nuclei, are typically used for the staining and counting of neuron perikaryons (Mullen, Buck & Smith, 1992).

This study successfully identified a collection of protocols for the characterization of cell populations in the histopathology of encephalitic diseases in equine CNS tissues. This suite included IHC staining of CD3+ T lymphocytes, CD79αcy+ B lymphocytes, MAC387+ macrophages and neutrophils, Iba-1+ microglia, GFAP+ astrocytes, and NF-H+ neurons in FFPE tissues. The suite was expanded with the inclusion of CD4+ and CD8+ T lymphocytes in fresh, frozen tissue sections. Useful application of these antibodies was supported with the characterization of neuropathology in diseased horses with clinical and experimental West Nile encephalitis, clinical equine protozoal myeloencephalitis, and, preliminarily, clinical Eastern equine encephalitis. Western blot analysis utilizing these antibodies should next be performed to support the specificity of their reactivity in the horse.

We would like to thank Dr. Gerry Shaw and Dr. W. Jake Streit for their consultation and loan of antibodies.

Additional Information and Declarations

Competing Interests

Author Contributions

Animal Ethics

Data Deposition

The authors declare that they have no competing interests.

Gretchen H. Delcambre conceived and designed the experiments, performed the experiments, analyzed the data, wrote the paper, prepared figures and/or tables, reviewed drafts of the paper.

Junjie Liu performed the experiments, analyzed the data, reviewed drafts of the paper.

Jenna M. Herrington performed the experiments.

Kelsey Vallario performed the experiments.

Maureen T. Long conceived and designed the experiments, performed the experiments, analyzed the data, contributed reagents/materials/analysis tools, wrote the paper, reviewed drafts of the paper.

The following information was supplied relating to ethical approvals (i.e., approving body and any reference numbers):

Archived equine tissues were utilized under the University of Florida (Gainesville, FL) Institutional Animal Care and Use Committee protocols #F077, #F093, #D163, and #4109.

The following information was supplied regarding data availability:

The research in this article did not generate any raw data.

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
