# Peer review of "Immunohistochemistry for the detection of neural and inflammatory cells in equine brain tissue"

_PeerJ, doi:10.7717/peerj.1601_

## Round 0.1 · original submission · Minor Revisions

This manuscript was well received by the two reviewers. Please consider their suggestions in your revised manuscript.

Reviewer 1 ·

Basic reporting

No comments

Experimental design

L 108 – can you clarify the type and power of the microwave used for this?

Validity of the findings

No comments

Additional comments

Review of the manuscript entitled, “Immunohistochemistry for the detection of neural and inflammatory cells in equine brain tissue”, by Delcambre et al. This manuscript describes the optimization and validation of several IHC markers for use in Equine brain.

Major comments:
None

Minor comments:
Line 233-234 – Why is this line about overnight incubations included? If we are looking for the best staining –we need to pick one and the point this comment is unclear.

Images – Annotation of the images could be useful to improved the interpretation by the inexperienced reviewer. Additionally, the one set of arrows present is so small that it is nearly impossible to see – these need to be made large.

Reviewer 2 ·

Basic reporting

The article generally meets basic reporting requirements.
-Use of the term "cassette" for the set of antibodies evaluated is inappropriate and misleading. Suggest using 'set' instead. (a collection of distinct elements having specific common properties)
-Remove FFPE acronym from abstract as it is not used again.
-This entire manuscript is very acronym heavy. To the point of making it non-nonsensical in some sections. If a list of acronyms are going to be provided at the beginning of the paper, why are they then also defined within the manuscript? Acronyms should not be used to start a sentence.
-City, State, Country information only needed for first citation of a company in M&M.
-Single sentences should not be stand alone paragraphs. Please correct this at lines 145 and 153.
- Lines 247-252 of discussion belong in M&M
- Some figures could be combined (1 & 2), (3, 4, 5, & 6)
- Figure legends could be a bit more descriptive so that the reader does not have to refer to the text of the paper to understand staining procedures shown.

Experimental design

No Comments

Validity of the findings

No comments

Additional comments

Methods were adequately complete. Experimental design included proper controls, which is much appreciated. Results were not over-interpreted. Methods and reagents described in this report will be of value to the research community for investigating CNS disease in horses and likely in other species as well.

---

## Round 0.2 · accepted · Accept

The manuscript has been improved according to the suggestions received.